# Zika Virus Infection Damages the Testes in Pubertal Common Squirrel Monkeys (*Saimiri collinsi*)

**DOI:** 10.3390/v15030615

**Published:** 2023-02-23

**Authors:** Gabriela da Costa Benchimol, Josye Bianca Santos, Ana Sophia da Costa Lopes, Karol Guimarães Oliveira, Eviny Sayuri Trindade Okada, Bianca Nascimento de Alcantara, Washington Luiz Assunção Pereira, Danuza Leite Leão, Ana Cristina Carneiro Martins, Liliane Almeida Carneiro, Aline Amaral Imbeloni, Sheila Tetsume Makiama, Luiz Paulo Printes Albarelli de Castro, Leandro Nassar Coutinho, Lívia Medeiros Neves Casseb, Pedro Fernando da Costa Vasconcelos, Sheyla Farhayldes Souza Domingues, Daniele Barbosa de Almeida Medeiros, Sarah Raphaella Rocha de Azevedo Scalercio

**Affiliations:** 1Department of Arbovirology and Hemorrhagic Fevers, Evandro Chagas Institute, Ananindeua 67030-000, Pará, Brazil; 2Postgraduate Program in Virology (PPGV), Evandro Chagas Institute, Ananindeua 67030-000, Pará, Brazil; 3Laboratory of Amazon Animal Biotechnology and Medicine (BIOMEDAM), Federal University of Pará, Castanhal 68740-970, Pará, Brazil; 4Postgraduate Program in Animal Reproduction in the Amazon (ReproAmazon), Federal University of Pará, Castanhal 68740-970, Pará, Brazil; 5National Primate Center, Ananindeua 67000-000, Pará, Brazil; 6Laboratory of Animal Pathology (LABOPAT), Institute of Health and Animal Production, Federal Rural University of the Amazon, Belém 66077-830, Pará, Brazil; 7Mamirauá Institute for Sustainable Development, Tefé 69553-225, Amazonas, Brazil; 8The Brazilian Institute for the Environment and Renewable Natural Resources, Belém 66087-441, Pará, Brazil; 9Department of Pathology, Center of Biologic and Health Sciences, State University of Pará, Belém 66050-540, Pará, Brazil; 10School of Veterinary Medicine, Federal University of Pará, Castanhal 68740-970, Pará, Brazil

**Keywords:** Zika virus, testes damage, squirrel monkeys

## Abstract

During the Zika virus (ZIKV) outbreak and after evidence of its sexual transmission was obtained, concerns arose about the impact of the adverse effects of ZIKV infection on human fertility. In this study, we evaluated the clinical-laboratory aspects and testicular histopathological patterns of pubertal squirrel monkeys (*Saimiri collinsi*) infected with ZIKV, analyzing the effects at different stages of infection. The susceptibility of *S. collinsi* to ZIKV infection was confirmed by laboratory tests, which detected viremia (mean 1.63 × 10^6^ RNA copies/µL) and IgM antibody induction. Reduced fecal testosterone levels, severe testicular atrophy and prolonged orchitis were observed throughout the experiment by ultrasound. At 21 dpi, testicular damage associated with ZIKV was confirmed by histopathological and immunohistochemical (IHC) analyses. Tubular retraction, the degeneration and necrosis of somatic and germ cells in the seminiferous tubules, the proliferation of interstitial cells and an inflammatory infiltrate were observed. ZIKV antigen was identified in the same cells where tissue injuries were observed. In conclusion, squirrel monkeys were found to be susceptible to the Asian variant of ZIKV, and this model enabled the identification of multifocal lesions in the seminiferous tubules of the infected group evaluated. These findings may suggest an impact of ZIKV infection on male fertility.

## 1. Introduction

Zika virus (ZIKV) is an arthropod-born virus of the genus *Flavivirus*, family *Flaviviridae* [1]. ZIKV became globally notorious in 2015 after an outbreak in Central and South America [2,3], when it was declared a “Public Health Emergency of International Concern” by the World Health Organization [2,4] and a correlation between virus infection and microcephaly was observed [2]. Currently, the virus can lead to a set of congenital alterations named congenital Zika virus syndrome (CZS) [2,5,6]. Following this evidence, persistent virus infection in the male gonads after viremia clearance in mice was also observed, resulting in multiple cell damage [7,8,9]. Nowadays, discussions are being held regarding how ZIKV threatens male fertility [10].

Experiments in mice [7,8,9,11,12,13] and nonhuman primates [14,15,16,17], as well as ex vivo and in vitro human testicular cell investigations [13,18,19,20,21], have confirmed the ability of this virus to infect and cause the degeneration and necrosis of multiple testicular cell lines in the interstitium and within the seminiferous tubules, emphasizing the tropism of the virus and its ability to use the male reproductive tract as a reservoir [22,23,24]. Sexual transmission of the virus has also been described in humans [24]. Evidence suggests that the male reproductive tract may act as a reservoir for the virus after acute infection [25], implying that sexual transmission by both symptomatic and asymptomatic individuals can occur [26,27]. We also know that the presence of ZIKV has been reported in *Callithrix* sp. and *Sapajus* sp., suggesting the participation of neotropical nonhuman primates (NHPs) in the wild cycle of this virus in both urban and peri-urban areas [28]. ZIKV sexual transmission and persistence in tissues of the reproductive system are characteristics of ZIKV that contribute to public health concerns in two ways: (I) by contributing to the spread of the virus without the aid of an arthropod vector [29]; and (II) through the possibility of causing male sterility, as observed in experimental models [7,8,9]. Widespread transmission of the virus can impact the population of several different wild species by interfering with the sexual development of these animals. However, most studies address possible effects in adults, with limited elucidation of the potential impacts in immature individuals. A detailed description of the pathogenesis of this virus, as well as the short- and long-term effects of the virus on reproductive capacity, require further investigations [24,25].

In biomedical research, neotropical NHPs have emerged as excellent choices for experimental models since the previously used models (old world primates) carry a high maintenance value and risk of extinction [30]. The squirrel monkey (*Saimiri* sp.) is already a suitable experimental model for the study of diseases such as malaria [30] and has been identified as an excellent candidate for ZIKV research [31,32,33].

In our research, we studied the effects of ZIKV at different stages of experimental infection on pubertal male monkeys (*S. collinsi*), a species that has been proven to be a suitable animal model for investigating this disease [31,33]. We show the clinical-laboratorial aspect and anatomopathological patterns of the genital tract of male squirrel monkeys infected by ZIKV. We described the clinical-laboratory aspects and anatomopathological patterns of the genital tract of squirrel monkey males infected by ZIKV. To introduce noninvasive methods for this approach, we evaluated testosterone in fecal extracts, the gonadosomatic index (biometric measurements), and testicular echogenicity to assess the effects at different stages of infection. In addition, we performed a histomorphometric evaluation and determined Johnsen’s score as complementary testicular histological evaluations of this infection to perform a quantitative analysis in the comparison between infected and noninfected animals.

## 2. Materials and Methods

### 2.1. Ethical and Legal Aspects

All the NHPs used in this experiment belong to the colony of the National Primate Center (CENP) of the Evandro Chagas Institute (IEC), in Ananindeua, Para, Brazil (latitude 1°38′26″, longitude 48°38′22″). This study was previously approved by the Animal Research Ethics Committee (CEUA) of the IEC (protocol number: CEUA/IEC—n° 033/2018) as well as by the System Authorization and Information on Biodiversity of the Chico Mendes Institute of Biodiversity of the Brazilian Environment Ministry (SISBIO/ICMBio—n° 52928-2).

The animals were housed in individual cages, with dimensions of 0.9 × 0.8 × 0.8 m (length, width, and height), under natural photoperiod (12 h of light and 12 h of dark). The local climate is tropical humid, with an average annual temperature of 28 °C. The diet of the animals consisted of fresh fruits, vegetables, commercial primate chow specific for neotropical nonhuman primates (MEGAZOO^®^ P18, Protein 18%, Max. fiber 6.5%, Betim, Minas Gerais, Brazil), vitamin and mineral supplements, and water ad libitum.

Aiming to minimize the suffering of the animals and ensure their welfare, we established alternate days for sample collection, and the establishment of non-invasive techniques, and we opted for physical restraint of the animals, disregarding the use of chemical methods due to the more difficult return of the animals.

### 2.2. Experimental Design

Five (*n* = 5) *S. collinsi* pubertal males were selected. These animals belonged to the breeding stock of CENP and were between one year and seven months old and two years and seven months old (1 y 7 m–2 y 7 m).

These animals were submitted to serological analyses using the hemagglutination inhibition (HI) technique, which was performed in the Arbovirology and Hemorrhagic Fevers Section of the IEC. Animals that presented negative results (<20) for the presence of antibodies against arboviruses, including those for Dengue and Yellow fever viruses, were selected for the study. Only animals that were determined to be healthy were included in the project.

Two animals (*n* = 2) were isolated to compose the control group (G1) and inoculated with a 0.5 mL suspension of strain-free VERO cells (ATCC number CCL-81, Washington, DC, USA). The other animals (G2; *n* = 3) were experimentally infected by the Asian genotype of ZIKV, strain BE H815744 (Genbank KU365780) isolated by Department of Arbovirology and Hemorrhagic Fevers, Evandro Chagas Institute, Ananindeua, Brazil, the inoculum with a 0.5 mL suspension of infected VERO cells containing 1.0 × 10^5^ PFU/mL via intradermal injection at the level of the third right intercostal space below the right nipple [34] (Figure 1).

At preestablished times (Day −5 to Day 21) after ZIKV inoculation, all animals in the experiment were monitored for different clinical parameters. Fecal collection was performed daily, starting 5 days before ZIKV inoculation (Days −5 to 21). Testicular biometry and a physical examination were performed, and blood was collected for laboratory tests on alternate days (Days 0, 3, 5, 7, 10, 14, and 21). Ultrasonographic examinations were performed weekly (Days 0, 7, 14, and 21) (Figure 1).

At 21 days postinfection (dpi), the animals were euthanized with prior anesthesia of ketamine hydrochloride (15 mg/kg) and xylazine hydrochloride (0.5 mg/kg) intramuscular injection, then animals were euthanized by an overdose of fentanyl (40 µg/kg) in ketamine hydrochloride (20 mg/kg) administered intravenously, following the recommendations of the National Committee for the Control of Animal Experimentation (CONCEA). Subsequently, the testicles were collected by necropsy, and histopathological analyses and immunohistochemical (IHC) viral detection were performed.

### 2.3. Testicular Evaluation and Clinical Parameters

The animals were placed in the dorsal decubitus position, and both testicles were measured for length (craniocaudal), width (mediolateral), and height (dorsoventral) using a universal pachymeter. The total testicular circumference was determined using a tape measure. The volume of the right (RTV) and left (LTV) testicles was obtained using the formula Testicular Volume (cm^3^) = Length × Height × Width × 0.524 [35].

Total testicular volume (TTV) was obtained from the sum of the RTV and LTV. For testicular weight (g) the formula *d = m*/*v* was used, where d corresponds to the general mammalian testicular density (1.046 ± 0.003 g) [36], *v* to the TTV, and *m* the weight in grams. The gonadosomatic index (GSI) (%) was calculated based on the equation testes weight/total body weight × 100 [37].

The animals were restrained and weighed with common weighing scales (CLINCK, São Paulo, Brazil). At preestablished times (Day 0 to Day 21) after ZIKV inoculation, all animals were observed twice per day by a trained team to recognize any pain signals and assess appetite and stool quality. In addition to changes in weight, rectal temperature and mucosal coloration, intestinal disorder, skin lesions, and activity patterns of the animals (depression, inactivity, or behavioral stereotype) were evaluated.

### 2.4. Fecal Testosterone Levels

Fecal sample collection was performed daily between 7:00 am and 11:00 am. The material was placed in sterile plastic containers with screw caps and immediately sent to the Laboratory Section of CENP. Samples were kept in a freezer at −70 °C (Ilshin Lab Co., Ltd., Seoul, Republic of Korea), where they remained before the extraction and evaluation of sex steroids levels.

The extraction of fecal metabolites was performed from a 0.48–0.52 g (0.50 g) aliquot of feces in natura added to 5 mL of 80% methanol. Then, the diluted samples were homogenized overnight and subjected to centrifugation at 3000 rpm for 15 min. The supernatant was stored in microtubes, which were labeled with the animal and date and stored at −70° C until the evaluation of hormone levels [38,39]. Fecal testosterone levels (ng/mL) were measured by a chemiluminescence assay using the VITROS^®^ immunoassay system (VITROS^®^ ECiQ, Ortho Clinical Diagnostics, NJ, USA). All protocols followed the manufacturer’s specifications.

### 2.5. RT-qPCR for ZIKV Genome Detection

A total of 200 µL of blood and 100 mg of testicular tissue of the infected animals were mixed separately with PBS pH 7.4 with a Tissuelyser (QIAGEN, Carlsbad, CA, USA) and subjected to viral RNA extraction using the TRIzol^®^ Plus RNA Purification Kit (Thermo Fisher Scientific, Carlsbad, CA, USA). The RT-qPCR protocol [40], using QuantiTect^®^ Probe PCR Kits (QIAGEN, Carlsbad, CA, USA), was performed with a 7500 Fast Real-time PCR System (Thermo Fisher Scientific). A standard curve with synthetic amplicon targets included in the PUC plasmid was used for RNA quantification [41]. The RNAse P gene was used as an endogenous internal control [42].

### 2.6. Mac-ELISA to IgM Detection

Enzyme-linked immunosorbent assays (ELISAs) were performed to detect IgM antibodies against ZIKV in the sera obtained from blood samples collected from the NHPs [33,43], using a polyclonal anti-monkey IgM (KPL, Milford, VT, USA) and flavivirus antibodies conjugated with peroxidase (6B6c-1). The optical density (O.D.) was read in a spectrophotometer (Biotek, Winooski, VT, USA) using a 450 nm filter with a cutoff of 0.2.

### 2.7. Ultrasonographic Examination

The animals were restrained with leather gloves and were offered fruits, cricket (*Zophobas morio*) larvae, and condensed milk to minimize stress during the ultrasonographic exams [44].

Each animal was placed in the dorsal decubitus position, and ultrasound gel was applied directly to the skin (Maxicor Produtos Médicos Ltd.a., Pinhais, Brazil) of the scrotum to ensure good contact with the transducer. Logic E ultrasound equipment (GE Medical Systems, Wuxi, China) was used in B mode with a linear multifrequency transducer (8–18 MHz). Images of the right and left testicle scans were obtained in the sagittal (longitudinal) and transverse planes. Each parameter was measured three times, employing arithmetic means for calculation. Subsequently, the images were analyzed with ImageJ^®^ software (National Institute of Mental Health, Bethesda, MD, USA) [45].

Testicular echogenicity was obtained from the grayscale of pixels (0–255), where the minimum numbering (0) equaled the black tone pixel and the maximum numbering (255) equaled the white tone pixel, as previously described [46,47]. The analyses were performed in the sagittal (longitudinal) plane. Each testis was divided into four quadrants. The echogenicity of each quadrant was defined by the mean gray tone, represented in the histogram. Echogenicity was averaged for each animal per day.

For every animal, D0 was established as a control day for the calculation of echogenicity, represented by the value of 100%. The echogenicity rate (%) of the remaining days (Days 7, 14, and 21) was calculated as a function of their control day, establishing a percentage (Echogenicity rate % = Day x/Day 0).

### 2.8. Histological Analysis and Immunohistochemical Analysis

Testes anatomical regions were evaluated by optical microscopy (Zeiss, Oberkochen, Germany) at 10 and 40× magnification by two veterinary pathologists, and histopathological analyses were performed on paraffinized tissue slides cut into 5 μm sections and stained with hematoxylin-eosin (HE).

The measurement of seminiferous tubules and the evaluation of histological patterns were performed with ImageJ^®^ software [45]. Ten tubules from each testicle were randomly selected for histomorphometric measurements. The tubular area (µm^2^), seminiferous epithelium area (µm^2^), epithelium height/thickness (µm), and larger and smaller diameter (µm) were measured [48]. Sertoli, Leydig, and spermatogenic lineage cells (spermatogonia, spermatocytes, spermatids, and spermatozoa) were identified and quantified. An adaptation of Johnsen’s score [49,50] was used to evaluate the histological pattern of the germinal epithelium (Table 1). The histopathological analyses were performed in three histological fields, as currently recommended by the International Harmonization of Toxicologic Pathology Nomenclature.

The avidin-biotin-peroxidase method was adapted to detect the viral antigen using an anti-ZIKV polyclonal antibody produced in mice at the IEC. Based on previous studies [51,52,53], the peroxidase method was used for tissue immunostaining with a specific monoclonal antibody. The slides were quantitatively analyzed by selecting 10 random fields using 40× magnification under light microscopy. Then, the mean number of cells was calculated, and the result was multiplied by 0.0625 mm^2^, referring to the graduated reticle of the optical microscope [52].

### 2.9. Statistical Analysis

All data were expressed as mean ± SD (standard deviation) or minimum and maximum value and analyzed by Minitab^®^ 19 Statistical Software (State College, PA, USA) and GraphPad Prism Software 9.3.1 (GraphPad Software, San Diego, CA, USA). The normality of the data was verified by the Kolmogorov–Smirnov test. For univariate analysis, frequencies and measures of centrality and dispersion were obtained. Two-way ANOVA with Bonferroni post-test was used for the IHC quantifications. Fecal testosterone levels, testicular volume, gonadossomatic index (GSI), ZIKV RNA in testes, and body weight were subjected to logarithmic transformation for normalization of the data before being subjected to statistical tests. For mean differences between uninfected and infected animals, Student’s *t*-test was performed. In the analysis by infection stages, the means were differentiated by Student’s *t*-test (for the two-stage analysis) and analysis of variance (ANOVA) (for the three-stage analysis), followed by Fisher’s test for a significant difference. The original values were used in the graphs. The infection stages are represented by the periods: preinoculation (day −5 to −1), acute phase (day 0 to 9), and convalescent phase (day 10 to 21). Means of the normalized data can be seen at the Appendix A. Level of significance adopted for all tests was *p* < 0.05 (significant *p* < 0.05, very significant *p* < 0.03, extremely significant *p* < 0.001).

## 3. Results

### 3.1. Clinical Evaluation of ZIKV Infection in Male Saimiri

In general, infected animals showed few clinical signs. The mucous membrane remained normal in all animals during the experiment. None of the animals showed bowel disorder, depression, inactivity, stereotypical behavior, significant weight loss or skin lesions (Table 2). Based on rectal temperature parameters of *S. collinsi* [33], body temperature changes were observed in the infected group that showed fever (average 39.5 °C) between 3 dpi and 10 dpi. It is also noteworthy that at 5 dpi, all infected animals were feverish; however, at 7 dpi, the highest temperatures (40.5 °C) were measured in animals AT-005 and AT-163 (Figure 2A).

### 3.2. Viremia and Specific Antibody Profile

The RNA viral load in the blood of infected animals was detected at 3 dpi, where the viremic peak was at 5 dpi (mean 5.73 × 10^6^ copies/µL RNA copies). For animals AT-163 and AT-005, viral RNA was still detected at 10 dpi (Figure 2B). Regarding the humoral immune response, IgM seroconversion was observed from 14 dpi for all individuals and persisted at 21 dpi (Figure 2C). These two results complement each other and provide us with information on the course of ZIKV infection. The viremia period was defined as the acute phase (0 to 9 dpi) and was associated with the fever curve (3 to 9 dpi). At 10 dpi, antibody production was detected, while viral replication and symptoms improved, thus starting the convalescent phase (Days 10 to 21).

### 3.3. Morphometric Evaluation (In Vivo)

The body weight difference between G1 and G2 was not statistically significant, and weight loss was not detected in the infected group (G2) (Figure 2D). However, G2 exhibited a significant testicular volume loss as well as a lower gonadosomatic index (relationship between animal body weight and testicular weight—GSI) than G1, which was observed in both the acute (*p* = 0.016/*p* = 0.035) and convalescent (*p* = 0.013/*p* = 0.011) phases (Figure 2E,F). Furthermore, G2 also showed a significant testicular volume loss (*p* = 0.011) and a GSI reduction (*p* = 0.042) between infection stages.

### 3.4. Fecal Testosterone Levels

The average fecal testosterone level was 532.7 ± 292.3 ng/dL for G1 and 473.5 ± 284.4 ng/dL for G2. There was no significant difference between G1 and G2 (Figure 2G). However, the intragroup evaluation showed a normal balance during the experiment for G1, while G2 showed a significant reduction in comparisons of the preinoculation and acute phases (*p* = 0.006) and the acute and convalescent phases (*p* = 0.006).

### 3.5. Testicular Echogenicity

The total testicular echogenicity was 0.8144 ± 0.207% and 0.8424 ± 0.199% for G1 and G2, respectively. Although subjective analysis of ultrasonographic images showed a slight difference between some animals in both groups (Figure 3A–C), no significant difference was observed between G1 and G2 (Figure 3D). However, G2 animals showed considerable testicular echogenicity loss in the convalescent phase (Figure 3D) compared with the acute phase (*p* = 0.010).

### 3.6. Morphometric Evaluation and Macroscopic Changes in the Testicles of ZIKV-Infected Animals

After necropsy, morphologically, the testes of G2 showed few changes. A subtle difference in organ volume was noticed between the groups (Figure 4A,B). These data became more evident in intragroup analysis of GSI, where AT-163 (0.55) and AT-156 (0.22) had the highest and lowest values postmortem, respectively (Figure 4C). Regarding tissue RNA viral load, animal AT-156 showed a higher viral RNA detection value (4.22 × 10^0^ RNA copies/mg) while animal AT-163 showed a lower value (2.30 × 10^−2^ RNA copies/mg) (Figure 4D). A very strong inverse correlation was observed between postmortem GSI and RNA viral load in the testes: R= −0.9986 and *p* = 0.033 (Figure 4E).

### 3.7. Histopathological Analysis of the Reproductive System of Young S. collinsi Infected with ZIKV

The histopathological analyses showed that in both control animals, the presence of spermatozoa was observed in the seminiferous tubules in the right testis (+++), although in the left testis (+), only animal AT-003 had gametes; non-significative injures were observed for those animals (Figure 5A,B). On the other hand, at 21 dpi, all infected animals presented different degrees of the degeneration and/or necrosis of the cells of the seminiferous tubules, in addition to an inflammatory process characterized by mononuclear cell infiltration (neutrophils, leukocytes, or lymphocytes) that was concentrated mainly around small- and medium-sized blood vessels (Figure 5C,D). No orchitis, hemorrhage or tissue regeneration changes were observed.

Regarding testes spermatogenesis, the infected NHP AT-005 showed a retraction of the seminiferous tubules, which was associated with a greater luminal desquamation and reduction of the germ cells number with more evidence of Sertoli cells. Also observed were (I) intense vacuolization and cellular swelling (degenerated cells), with increased interstitial cellularity; (II) mild to moderate cellular necrosis; (III) mild to moderate interstitial edema and seminiferous tubule displacement from the basement membrane, suggesting an atrophic process; and (IV) some areas in the left testis with multifocal presence of lymphocytes (mild inflammation). No SPZT was observed in the tubular lumen (Figure 5E,F).

In animal AT-156, both testes showed decreased spermatogenic cell numbers with greater evidence of Sertoli cells. Intense degeneration of the tubular parenchyma was observed in areas with cellular swelling, vacuolization, and necrosis, but without rupture of the basal laminae. Multifocal interstitial edema and moderate tubular retraction were observed in the right testis, accompanied by an inflammatory filtrate, ranging from mild to moderate intensity, characterized by lymphocytes and neutrophils. No SPZT were observed in the tubular lumen.

The testes of animal AT-163 showed (I) marked and diffuse hydropic degeneration in the spermatogenic cells, which was more pronounced in the right organ than the left; (II) a decrease in the number of spermatogenic cells, with greater evidence of Sertoli cells; and (III) an absence of SPZT in the tubular lumen, even though this animal was of the same average age (2 years and 4 months) as the controls (~2.5 years) that had SPZT. Some tubules showed a lumen loss and moderate tubular retraction, compromising the seminiferous tubule architecture. In the interstitium, a mild inflammatory infiltrate was observed, predominantly lymphocytic and multifocally distributed, in addition to mild edema.

In histomorphometric analyses, G1 and G2 showed a significant difference in all parameters evaluated in the histological analysis at 21 dpi. G1 presented seminiferous tubules with a medium to high Johnsen score, with the presence of cells in an advanced process of differentiation, including the appearance of round and elongated spermatids. The mean values obtained in G2 were lower than those in G1 for tubular area (µm^2^) (*p* = 0.000), seminiferous epithelium (µm^2^) (*p* = 0.000), larger (*p* = 0.000) and smaller (*p* = 0.000) diameter (µm), epithelium height (µm) (*p* = 0.000), and Johnsen’s score (*p* = 0.000). In the Johnsen score analysis, a considerable number of tubules with a total absence of spermatocytes and spermatids and a reduction in the number of spermatogonia were observed (Table 3).

### 3.8. Immunohistochemistry for Viral Antigen Detection

The presence of the antigens for ZIKV was confirmed by IHC in the testes of all infected animals, confirming the tissue tropism of ZIKV in the testes of young *S. collinsi* males. Staining was observed in both vacuolized and/or pyknotic sperm cells and Sertoli and Leydig cells (Figure 5G).

The quantification of the viral antigen was observed only in the infected group (Figure 5H). Animals AT-163 and AT-005 showed a difference in marking between the right and left testes (Figure 5I), with more stained cells in the right testis, while in animal AT-156 the number of cells showing viral antigen staining was equivalent between testes. Similarly, the right testis of animals AT-163 and AT-005 showed more cells that stained for ZIKV than that of animal AT-156. The quantification showed an average of 2.83 × 10^2^ cells/mm².

## 4. Discussion

Scientific knowledge about the routes of viral infection in the male reproductive tract by viral agents is still limited, even though there has been an increase in interest in this field since the ZIKV outbreak in 2015–2016 as a result of the collapse of the previous dogma regarding sexual transmission [54,55,56] and the identification of an association of arboviruses, in particular flaviviruses, with congenital alterations [6,57,58]. Furthermore, ZIKV has been shown to persist in the male reproductive system for at least six months and is detected in the semen of patients after acute infection [59].

It is certainly true that in immunodeficient mice, ZIKV is associated with chronic inflammation of the testes and epididymis, with a substantial reduction in sperm count and fertility [8,9,60]. This is because of the complete destruction of testicular morphology and the loss of peritubular myoid cells and spermatogonia [9].

Although immunodeficient mice are the most commonly used models because of their ease of handling, breeding, and housing, as well as other benefits, we must keep in mind that they are not natural hosts of ZIKV, and their immune system inhibits viral replication, making IFN manipulation necessary. Therefore, the potential of these studies and the extrapolation of results obtained with these animals to more complex organisms, such as NHP and humans, are limited [61,62]. However, in the face of so many results, doubt persists: In humans, would it be possible for us to observe the same deleterious effects on male fertility after infection with ZIKV?

Approximately 30 viral species can impair male fertility, are sexually transmitted and can even affect progeny [63]. One of the most current examples that can be cited is the etiological agent of the COVID-19 pandemic, SARS-CoV-2. This capability was demonstrated in a previous human case study performed postmortem [64]. The testicular lesions caused by SARS-CoV-2 included a combination of orchitis, vascular changes, basement membrane thickening, and reduced spermatogenesis associated with local infection, along with a mononuclear inflammatory infiltrate.

In search of an animal model closer to the human species, NHP emerge as a more adequate proposal due to presenting a robust adaptive immune system with a response comparable to the human one. Experimental studies have demonstrated the susceptibility of several species of NHP to ZIKV infection, characterized by: (I) clinical signs; (II) detection in the blood (viremia), saliva, urine, CSF, and semen; (III) induction of humoral response; (IV) and viral clearance in the testes, accompanied by inflammation [65,66,67].

In our study, to observe the extent of testicular injuries in sexually immature animals and considering the predilection of ZIKV for pluripotent cells, we selected *S. collinsi* pubertal males as a model. This species was chosen because these neotropical primates are scientifically recognized for being good experimental animal models in different biomedical research guidelines; for having susceptibility to infection with arboviruses, including ZIKV [31,33]; and for being available in the Amazon region [68,69].

To accomplish this, the characteristics of ZIKV infection were mimicked and the hypothesis that neotropical NHPs could serve as reservoirs and amplifiers of the wild cycle of ZIKV was tested [28,33,70].

The infecting dose of 10⁵ PFU/mL and the intradermal route proved efficient for the induction of the infection, as it produced viremia, a humoral immune response, and testicular lesions in the infected group, corroborating what has been verified in previous studies of other NHP species in which the same inoculation parameters cited were used [65,71].

The analysis of different stages of infection highlighted the abnormalities noted in this experiment. The difference in stages of infection is understood by the immunological reaction to ZIKV, which differs between the acute phase (viral replication) and the convalescent phase (the recovery phase associated with antibody production).

A reduction in testosterone levels in the acute phase compared to the preinoculation and convalescent phases was noted, presumably owing to damage to Leydig cells. Similar findings have been described in other studies with immunosuppressed mice [7,8,9]. We observed a recovery in testosterone production after the acute phase of infection. This implies that after the viral replication stage, there is regeneration of Leydig cells and a stabilization of androgen production. Adequate intratubular testosterone levels are essential for germ cell differentiation and Sertoli cell maturation, as well as for helping to maintain the blood–testicular barrier [72,73]. In adult rats, a detachment of the seminiferous epithelium was described, with subsequent apoptosis of germ cells after experimental testosterone withdrawal [74]. At the 21st dpi, we observed a detachment of the germinative epithelium in a significant proportion of the tubules in G2. We hypothesize that one of the reasons for this disturbance is a decrease in testosterone production.

The testicular volume data of all animals evaluated in this study, as well as their body weight [35,75,76,77], were considerably lower than those observed in sexually matured *S. collinsi* [35,75,76,77]. In G2, the correlation of testosterone levels and testicular volume contributed to the reduction in the GSI, and histomorphometry evaluation indicated a loss of scrotal mass, suggesting an immediate effect of androgen even though its normal production recovered by the end of the experiment. Comparable results have already been described in experimental studies with immunodeficient mice, with experimental ZIKV infection resulting in testicular involution, characterized by a reduced size and weight of the testicles [7,8,9].

Our study also points to a local inflammatory response persisting in the testicular parenchyma until the convalescent phase, as seen in the ultrasonographic and immuno-histochemistry evaluations. The persistence of the virus within the seminiferous tubules and the interstitium at 21 dpi has already been observed, associated with a mononuclear infiltrate around blood vessels. Significant echogenicity loss in the scrotal mass was also evidence of an inflammatory response, which was verified by both increased blood flow and edema.

Due to their immaturity, the testes of young KO A129 mice showed greater tissue damage than those of adult animals [60], in which no atrophy, inflammation, or hemorrhage was observed. Similar findings were described in a study of 51 NHPs (*Macaca mulatta* and *Macaca fascicularis*) from a rather heterogeneous sample group, in which higher rates of ZIKV RNA detection and histopathological lesions were observed in mature primates than sexually immature primates; however, these findings were observed more in the epididymis, prostate, and seminal vesicles than the testes, which were not available or samples were excluded [14]. In contrast to the aforementioned research, in our study, it was possible to observe histopathological damage and to detect ZIKV RNA in pubertal animals. However, our study focused on testicular damage and not on the entire male genital tract, similar to the research performed in KO A129 mice.

It was possible to demonstrate that pubertal *S. collinsi* showed testicular tissue changes at 21 dpi with ZIKV, similar to what was observed in the experiments [60] using infected young IFN-I receptor-deficient mice (strain A129). However, the lesions found in this work were of milder severity, and animals showed no signs of hemorrhage, regeneration, or atrophy of the organ [9]. Although the organ collection time point was at 21 dpi for both studies, the difference in lesion severity between *S. collinsi* and A129 knockout mice may be explained by the maturation time of the sex organs, which in the case of mice occurs in a shorter period. Therefore, it is recommended to perform kinetics analyses with the monkeys, with multiple collections at longer infection times. Through these analyses, we could obtain in-depth results and observe the establishment of irreversible lesions, which could lead to infertility or the complete regeneration of the organ.

The experimental design presented here made the following possible:

An analysis of the kinetics of infection, with evidence of fever (Figure 2A), viremia (Figure 2B), and IgM antibody detection (Figure 2C). Thus, it was possible to delineate the acute phase of infection (between 1 and 10 dpi) and the onset of the convalescent phase (10 dpi to 21 dpi), periods that are similar to those in other research with NHPs as well as humans [14,78]. Interestingly, animal AT-005 showed a persistent febrile state between 3 and 10 dpi, which was observed to be consistent with the viremic period (3 to 10 dpi), and a higher O.D. (0.38) of IgM at 14 dpi (Figure 2C). Proportionally, NHP AT-163, which was no longer feverish as early as 10 dpi, had the shortest viremic period and had the lowest IgM titers.

The GSI helps in determining the reproductive cycle since the maturation of reproductive cells coincides with the enlargement of the gonads as the animal develops [79]. It is worth noting that there is an inversely proportional correlation (R= −0.9986; Figure 4E) between GSI (Figure 4C) and viral load in the testes (Figure 4D); for example, the animal with lower GSI tends to have a higher viral load in the tissue, while the animal with higher GSI has a lower tissue RNA quantification, most likely due to its higher gonad development.

In the microscopic analysis, the histopathological injuries found in this study were limited to the presence of inflammatory infiltrate, tubular retraction associated with peritubular edema and/or interstitial cell proliferation, and the cytoplasmic vacuolization and/or pyknosis of germ cells of the seminiferous tubule (Figure 5). We highlight the following significant results: (I) the reduction in the number of spermatogenic cells, (II) the increase in the number of Sertoli cells (Figure 5D), and (III) the rupture of the basement membranes of the seminiferous tubules. Such injury was so extensive that some tubules were almost completely retracted (Figure 5C), showing signs of tissue atrophy. Despite the testicle being an immune-privileged region, the intricate defense system of individuals in puberty is still developing [80,81].

Furthermore, no signs of regeneration were seen, since even after 21 dpi, the tissue was still inflamed. Again, an evaluation with a longer period of exposure to the virus may make it possible to observe the outcomes of ZIKV infection.

In ZIKV-infected adult olive baboons (*Papio anubis*), decreased spermatogenesis was observed in certain individuals, resulting in oligospermia and aspermia. However, the authors reported no major testicular pathology in the remaining animals [16]. Similar outcomes have been found in rhesus monkeys, with reduced germinal epithelium in certain seminiferous tubules but no significant pathological abnormalities in the experimentally infected animals [15]. The histological evaluation in our study revealed more severe alterations than those already described in these NHPs. The severity of the findings may be directly related to an agent-specific immune strategy that differs between old world and new world monkeys, emphasizing the significance of selecting the most appropriate experimental model for the investigation of a particular disease [82,83,84,85]. Additionally, recalling the similarity of the results reported here with what has already been described in immunosuppressed rodents [7,8,9], animals of the *S. collinsi* species may have a greater susceptibility to ZIKV infection than other NHP species, as already published [33].

Decreased spermatogenic cells in our findings can be clarified by the overcoming of the blood–testicular barrier (BTB), as noted by other authors [9,16,22,86]. The BTB is an immune-privileged barrier critical in maintaining spermatogenesis and is formed by tight junctions between Sertoli cells. Sertoli cells can modulate the immune response by regulating immunoregulatory factors, protecting the mature spermatogenic cells present in the compartment from immune action [87]. As noted in this study, the virus detection within the tubules confirms the ability of the virus to overcome the BTB. Human in vitro studies have clarified that ZIKV does not cross the barrier by downregulating the expression of junction proteins but rather by infecting macrophages, which then secrete inflammatory mediators, thus disrupting homeostasis [19,22]. The infection of Sertoli cells is crucial to the antiviral response and ZIKV replication, explaining the persistence of the virus inside the tubules [8,12,13,19,20,21,86].

Additionally, Sertoli cells are necessary for the nutrition and support of germ cells in development and are responsible for the production of inhibin B, a fundamental protein in the continuity of the spermatogenic process that is directly correlated with testicular volume [88,89]. ZIKV infection downregulates Sertoli cells, adversely affecting signaling pathways important in spermatogenesis and negatively affecting inhibin B expression [22,90]. Furthermore, the persistence of ZIKV RNA in testicular tissue in rodents [62,91] as well as in NHPs [17,67,84] has been reported, confirming enduring ongoing viral aggression in the germinative epithelium. A long cascade composed of Leydig and Sertoli cell infection, reduced hormonal production, and cell degeneration and death directly through viral lesions contributes to testicular atrophy. This set of abnormalities demonstrates the threat of this virus to male fertility and draws attention to the need for prophylactic methods such as vaccines.

There is still no histological description of the testes of *S. collinsi* juvenile or adult animals, nor are there any studies defining the exact time at which these animals reach puberty and sexual maturation. Puberty is a temporal process characterized by the transition to adulthood and the sexual development of the individual so that it is capable of reproduction. In the male, this process is marked by a cascade of morphological changes, including the growth and maturation of the gonads, associated with increased secretion of sex steroids and the initiation of spermatogenesis [81,92]. The presence of spermatozoa in the seminiferous tubules of these animals was observed in this study, with the presence of cells in an advanced differentiation process. These findings suggest that the spermatogenic process starts before two years and two months, the average age of the animals used in this study. It is worth mentioning that only the control group presented spermatozoa in its tubular lumen and that, in the infected animals of the same age, these cells were not detected. However, the difference in measurements such as testicular volume compared to adult animals suggests that the animals in this study were not yet fully mature. Therefore, we cannot point out whether these animals are already fertile. Studies to elucidate the precise age at which these animals are capable of producing mature sperm and reproducing are needed. Thus, it would be interesting to follow young infected animals until they reach complete sexual maturity to evaluate the deleterious effects of ZIKV on the fertility of this species.

In the same way, one should also consider a very relevant question in the case of captive animals: Do these animals, by having better environmental and feeding conditions, sexually mature earlier than animals in the wild? This question is directly associated with the fact that in G1, with an average age of 2.5 years, SPTZ could be detected in the lumen of the seminiferous tubules. We leave this question as a topic for future experiments.

As pointed out by Ball et al. (2022) [14], formalin fixation damages testicular tissue, making histological evaluation difficult. Consequently, we adopted Johnsen’s score to better evaluate the degree of spermatogenic cell differentiation. Such methodology presented itself as an excellent choice for the purpose of this work, offering a quantitative evaluation of the results. Similarly, the computed-assisted testicular echogenicity evaluation introduced a new way to evaluate the testicular parenchyma. Both methods, analyzed by the open-source software ImageJ, are described here as accessible methodologies for future studies.

Our work not only presents *S. collinsi* as a suitable experimental model for biomedical research investigating the effects of Zika virus on the male reproductive tract but also warns about the potential transmission of ZIKV in this free-living species. Even though *S. collinsi* is classified as species of “least concern” by the International Union for Conservation of Nature’s Red List of Threatened Species [93], the high severity of the findings in this study raises alarm regarding the possibility of the virus affecting the population of these animals in the wild, particularly due to the persistence of the local inflammatory response even after the viremic phase has ceased.

In summary, neotropical NHPs of the species *S. collinsi* are susceptible to the Asian variant of ZIKV, and ZIKV infection leads to a spectrum of histopathological changes at the testicular level in prepubertal males, characterized by the degeneration and/or death of germ cells and Sertoli cells, accompanied by inflammation and the proliferation of Leydig cells. Furthermore, these findings reiterate the potential for ZIKV invasion into the male reproductive system and suggest a risk to animal fertility.

## 5. Conclusions

Here, we present important results demonstrating multiple effects of experimental infection with the Asian ZIKV strain. Reduced testosterone levels, testicular atrophy and a prolonged inflammatory process were observed. We validated the feasibility of the use of noninvasive methods such as fecal extract analysis to evaluate testosterone levels and testicular ultrasonography to study these effects. In addition, testicular histomorphometry proved to be an excellent methodology for accurate measurement and evaluation of the germinal epithelium of the animals studied. The study of *S. collinsi* showed us that this species is highly susceptible to ZIKV, as evidenced by the febrile state, viremia, humoral response induction, and viral testicular immunolocalization, even allowing the evaluation of a correlation between the reduction of testicular volume and the increase of viral load in the testes, raising awareness for the high risk of infection in free-living animals and their possible role as a virus reservoir. Furthermore, these animals are an interesting experimental model to study fertility disorders caused by ZIKV. Further studies evaluating the long-term effects of ZIKV infection, exploring the exact mechanisms of viral pathogenesis and its immune reaction, and addressing the effects of the virus on fertility rates in adult animals are needed.

## Figures and Tables

**Figure 1 viruses-15-00615-f001:**
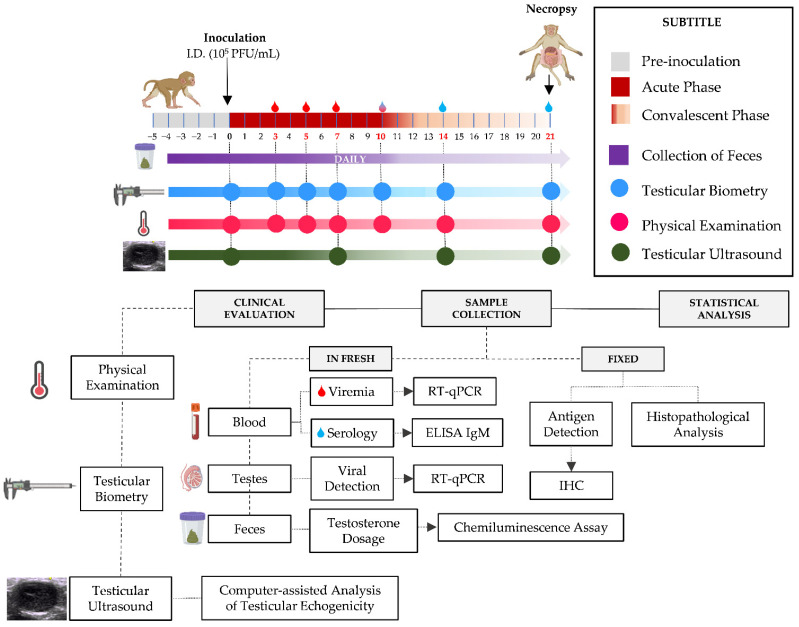
Experimental design of the study. The upper portion of the figure shows the timeline of sample collection and biometric data from the animals evaluated; the periods were divided into (1) Pre-inoculation, (2) Acute and (3) Convalescence phases of infection, each with its corresponding color according to the legend on the right of the figure; on the ruler we can see the days marked in red on which biological material was collected, the day of inoculation and euthanasia of the animals, as well as what was collected and evaluated, and on which days they were evaluated. In the lower portion of the figure, we have a flowchart showing to which methodology each of the samples was forwarded and the culmination of the entire study.

**Figure 2 viruses-15-00615-f002:**
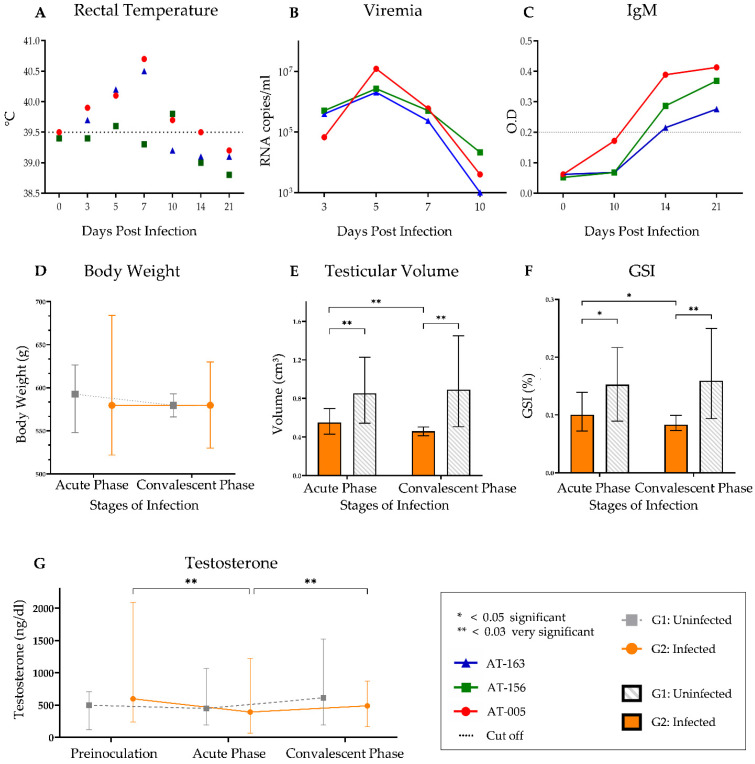
Noninvasive evaluation of experimental ZIKA infection in male *S. collinsi*. (**A**) Measurement of rectal temperature, showing feverish period of infected group between 3 to 10 dpi. The dashed line represents the temperature cut-off based in control group data. The (**B**) RNA viral load in blood by RT-qPCR and (**C**) IgM antibody profile of the infected group detected from 14 dpi. The dashed line represents the cut-off of ELISA assay. Group evaluation, mean and min-max values: (**D**) body weight, (**E**) testicular volume, (**F**) gonadosomatic index, and (**G**) testosterone by stages of infection, highlighting the decrease of testicular volume, GSI and testosterone values during acute phase of infection in G1.

**Figure 3 viruses-15-00615-f003:**
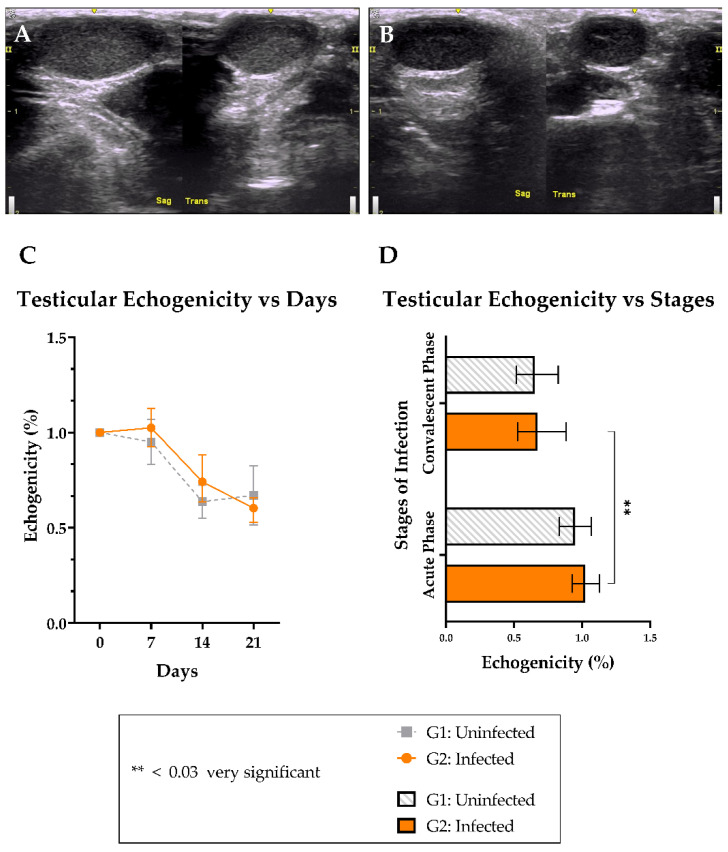
Ultrasonographic analysis at 21 dpi. The left testicle of animal AT-003 (G1) showed normal homogeneity (**A**) and the left testicle of animal AT-163 (G2) presented hypoechoicity in all its extensions (**B**). The testicular echogenicity of G1 and G2 was evaluated at the indicated day (**C**) and by stages of ZIKV infection (**D**). Mean and min-max values.

**Figure 4 viruses-15-00615-f004:**
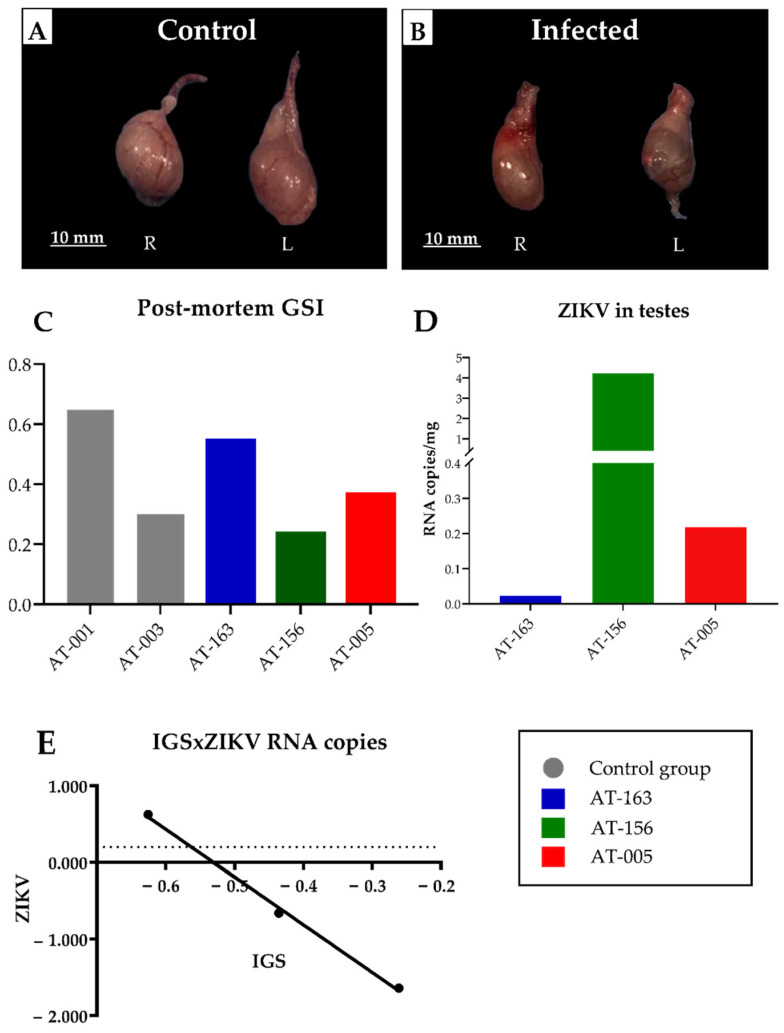
Postmortem evaluation of experimental ZIKA infection in male *S. collinsi*. (**A**) Control animal AT-001, macroscopy of the right and left testes. (**B**) Infected animal AT-156, macroscopy showing a slight difference in size between the right and left testes. (**C**) Postmortem GSI graph at 21 dpi of each animal. (**D**) Graph of the detection of ZIKV RNA copies/mg in a pool of both testes of each infected animal. (**E**) Graph of Pearson’s correlation: demonstrating a significant inverse and very strong correlation between GSI X ZIKV tissue variables, R-value = −0.9986 and *p* = 0.033.

**Figure 5 viruses-15-00615-f005:**
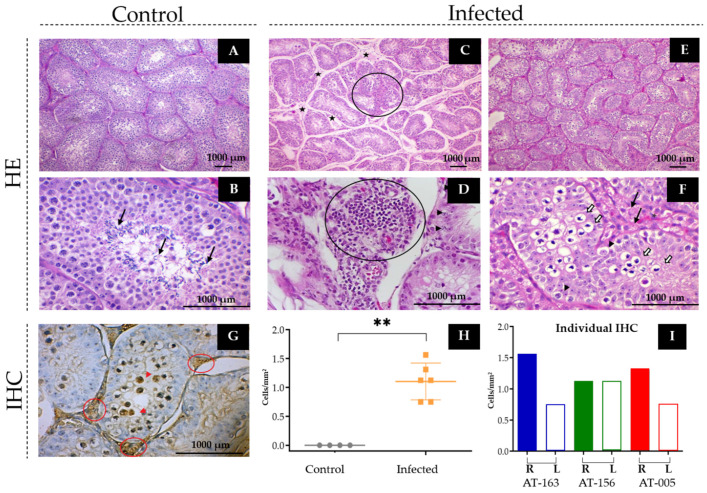
Microscopic evaluation of *S. collinsi*. (**A**) Integrity of testicular tissue from control animal AT-001 (100×). (**B**) Seminiferous tubule of the control animal, in a higher magnification lens, showing sperm (arrow), score 10 (400×). (**C**) AT-005, infected with ZIKV, showing diffuse tubular edema and tubular retraction (star). Highlighted (circle) interstitial cell proliferation associated with inflammatory infiltrate (100×). (**D**) Perivascular inflammatory infiltrate (circle), with the presence of lymphocytes and neutrophils. The seminiferous tubule of an infected animal, showing a decrease in the number of germ cells, which are degenerated (arrowhead) (400×). (**E**) AT-163 infected animal, showing proliferation of Leydig cells and degenerated germ cells. (**F**) Infected animal, showing seminiferous tubule containing degenerating spermatogenic cells (hollow arrows), presence of pyknotic nuclei and cytoplasmic macrovacuolization (arrowhead), score 4. In the extra tubular area, there is an increase in the proliferation of Leydig cells (black arrows) (400×). (**G**) Light micrograph showing IHC staining with peroxidase using anti-ZIKV polyclonal antibodies. Seminiferous tubule from infected animal AT-005, showing ZIKV antigen labeling on both the germ cells of the seminiferous tubule (arrow) and the Leydig cells (circle). (**H**) Graph showing the quantification of viral antigen per group in the testicular tissue of the control and infected animals. ** *p* < 0.03. (**I**) Graph showing the quantification of ZIKV antigen per animal based on the IHC test. In blue is the animal AT-163, in green the AT-156 and in red the AT-005; the filled columns represent the right testis and the empty columns the left testis.

**Table 1 viruses-15-00615-t001:** Johnsen’s score for squirrel monkeys (*Saimiri collinsi*). Adapted from Johnsen (1970) and Manabe, Takeshima, and Akaza (1997).

Score	Histological Pattern
10	Fully mature spermatogenesis is observed (mature hooked spermatozoa with dense nuclear chromatin lying within tubular lumen).
9	Many spermatozoa are present, but the germinal epithelium shows markedsloughing or obliteration of the lumen.
8	Only a few spermatozoa (<10) are present in the section.
7	No spermatozoa but many mature spermatids are present.
6	Few mid-phase spermatids (<10), with pale chromatin and narrow oval heads, are observed; they are radially arranged.
5	Immature spermatids, randomly arranged throughout the tubule are observed; each has a rounded nucleus with pale chromatin.
4	Only a few spermatocytes are observed and no spermatids or spermatozoa.
3	Spermatogonia are the only germ cells present.
2	No germ cells are present, but Sertoli cells are.
1	No cells are present in the tubular section.

**Table 2 viruses-15-00615-t002:** Overview of the clinical evaluation of the animals used in the study.

Groups	ID	Age (Years)	Weight (g) Mean ± SD	Mucous
**G1**	AT-001	2.7	588 ± 19	normal colored
	AT-003	2.2	578 ± 6	normal colored
**G2**	AT-005	1.7	584 ± 12	normal colored
	AT-156	1.7	536 ± 8	normal colored
	AT-163	2.4	588 ± 44	normal colored

G1 = Control group; G2 = Infected group; ID = Animal identifier.

**Table 3 viruses-15-00615-t003:** Seminiferous tubule (*n* = 10 per testicle) measurements and Johnsen’s score of squirrel monkeys (*Saimiri collinsi*) on the 21st day following infection with Zika virus.

Groups	Groups	
	G1: Uninfected	G2: Infected
Tubular Area (µm^2^)	21 292 ± 7 334 ^a^	11 475 ± 2 373 ^b^
Seminiferous Epithelium Area (µm^2^)	15 983 ± 6 568 ^a^	7 961 ± 1 741 ^b^
Larger Diameter (µm)	179.6 ± 39.8 ^a^	131.2 ± 20.7 ^b^
Smaller Diameter (µm)	148.1 ± 27.3 ^a^	109.5 ± 10.2 ^b^
Epithelium Height (µm)	45.7 ± 13.5 ^a^	30.6 ± 4.8 ^b^
Johnsen’s Score	4.9 ± 1.05 ^a^	3.25 ± 0.47 ^b^

^a,b^ Different lowercase letters indicate significant differences between means comparing the experimental groups (*p* < 0.05).

## Data Availability

Not applicable.

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
