# Peer review of "Zika Virus Infection Damages the Testes in Pubertal Common Squirrel Monkeys (Saimiri collinsi)"

_viruses, 2023, doi:10.3390/v15030615_

Round 1

Reviewer 1 Report

In this study, the authors established sexually immature Squirrel Monkeys models of ZIKV infection to observe the extent of testicular injuries and demonstrated Neotropical NHP could serve as reservoirs and amplifiers of the wild cycle of ZIKV. The authors addressed these key questions in the models and warned that ZIKV infection might establish a persistent infection. However, several questions need to be clarified before this manuscript could be formally accepted.

1.     Figure 1A, what are the rectal temperatures of healthy group? Did the infected animals become feverish on the day when they were inoculated with the virus?

2.     Line 248 and Figure 1B, we usually describe the viral RNA level as eg.,5,73x106copies/ul(in blood samples)or /ug(in tissue samples). What did the authors mean by 5,73x106ng/µL?

3.     Line 252 “The viremia period was defined as the acute phase (0 to 10 dpi)...”but RNA viral load in the blood of infected animals was detected at 3 dpi (line 247). There seems to be a misstatement of the acute phase.

4.     Figure1C, the healthy controls are missing.

5.     Figure1G, why the authors examined the fecal testosterone level instead of more commonly used blood testosterone level? Besides, the fecal testosterone level varies too much in the infected group. Please make sure it meet the statistical requirements. More samples should be included. In this study, the author defined the 0-10 dpi as acute phase and 10-21 dpi as convalescent phase. It was not clearly stated whether the results shown in Figure 1G were the average of the 11 days in each phase. If there was a peak during the 11 days it might make a difference. As the authors mentioned the feces were collected daily, it would be better to measure the levels at shorter intervals.

6.     The existence of infectious viral particles in the testis of infected group on Day 21 should be examined by additional assays, such as Plague assays.

7.     Table1, it would be better to use “histological pattern” than “Title 3”.

8.     To be consistent with the notification in Figure 2 and Figure 3, where “P < 0.05 is considered as significant, P < 0.03 is considered as very significant, P < 0.001 is considered as highly significant”. Please correct the statement in Line232.

Reviewer 2 Report

Please, find the pdf file.

Round 2

Reviewer 2 Report

General Comment: Overall, the revised manuscript has greatly improved and most of major issues are addressed. However, I strongly suggest a professional English editing for the manuscript prior to the acceptance. The title, for instance, “of” should be omitted after the adjective “pubertal.”  

Specific comment:

- Line 355: To support this notion, the expression of testicular germ markers should be evaluated either by IHC, semi-quantitative western blot, or quantitative approaches.

AUTHORS´ REPLY:  We agree that will be interesting to perform the IHC to mark specific cells and quantify. However, we had restriction of financial support.

- Involution of the seminiferous tubules was observed in the histological section as it has been reported by others. Is it also followed by the loss/impairment of blood testis barrier?

AUTHORS´ REPLY:  The BTB is formed by Sertoli cells junctions, as the infection interferes with the architecture of the tubules, consequently they are compromise.

Regarding these issues, the authors should tone down their statements due to lack of direct indicators and (semi)quantitative data to avoid overinterpretation/overclaim

Author Response

Dear Reviewer 2

Thank you for your considation, but could you be more clear where we are overinterpretation/overclaim ? We change 3 point in the text (highlight in blue). I hope that we  may clarify your doubts. About the professional English editing, we sent in the previous submission, the certification that it was made. 

My best

Daniele Medeiros
